# Farmers' Preferences and Agronomic Evaluation of Dynamic Mixtures of Rice and Bean in Nepal

Shree Prasad Neupane [1,*], Bal Krishna Joshi [2], Dipendra Kumar Ayer [3], Krishna Hari Ghimire [2], Devendra Gauchan [4], Ajaya Karkee [2], Devra I. Jarvis [5,6], Dejene K. Mengistu [7], Stefania Grando [8,*] and Salvatore Ceccarelli [8,*]

[1]   Local Initiatives for Biodiversity, Research and Development (LI-BIRD), Pokhara P.O. Box 324, Nepal
[2]   National Genebank, Nepal Agricultural Research Council, Khumaltar, Lalitpur 44702, Nepal;
      bk.joshi@narc.gov.np (B.K.J.); ghimirekh@gmail.com (K.H.G.); ajayakarkee@gmail.com (A.K.)
[3]   Gauradaha Agriculture Campus, Tribhuvan University, Kathmandu 446000, Nepal; dip_ayer@hotmail.com
[4]   Bioversity International, Country Office, Kathmandu 446000, Nepal; d.gauchan@cgiar.org
[5]   Platform for Agrobiodiversity Research (PAR), c/o the Raffaella Foundation, Nordland, WA 98358, USA;
      d.jarvis@agrobiodiversitypar.org
[6]   Department of Crop and Soil Sciences, Washington State University, Pullman, WA 99164, USA
[7]   Bioversity International, ILRI Campus, Addis Ababa P.O. Box 5689, Ethiopia; d.mengistu@cgiar.org
[8]   Consultant, 63100 Ascoli Piceno, Italy
*    Correspondence: shree.neupane@libird.org (S.P.N.); sgrando56@gmail.com (S.G.);
     ceccarelli.salvatore83@gmail.com (S.C.)

**Abstract:** Field trials of rice and bean dynamic mixtures were carried out in low input and hill farming systems of Nepal from 2019 to 2021 to improve productivity and resilience. The rice trials were conducted in two locations (Jumla and Lamjung) and those on bean in Jumla, using a randomized complete block design with three replications. Dynamic mixtures were constructed from landraces, improved varieties and breeding lines for both crops. A total of 48 bean entries were used in Jumla, whereas 56 and 66 rice entries were used to make location-specific dynamic mixtures in Lamjung and Jumla, respectively. They were formed by mixing diverse varieties as a strategy to maintain a broad genetic base. Farmers (men and women) and technicians selected from the most complex mixture and the selections were added to the trials starting from second year. In rice, some mixtures and selections from the mixtures gave grain yield comparable to the improved check and higher than the local checks. In the case of bean, differences between entries were not significant but some of the selections received a high preference score. Overall, the dynamic mixtures appear as a reliable material for sustainable increase in yield in the low input and hill farming system of Nepal.

**Keywords:** dynamic mixtures; productivity; resilience; broad genetic base; entry x year interaction; farmers' preferences

## 1. Introduction

Agricultural intensification is a major cause of the narrowing of both spatial and temporal diversity within agricultural systems with the result of a decrease in crop diversity within fields and across landscapes [1]. The decrease in biodiversity in general, and of agrobiodiversity in particular, has a number of consequences such as a reduced resilience to climate change of the production systems [2]. Furthermore, agricultural practices such as high applications of fertilizers and pesticides can reduce the ability of ecosystems to provide goods and services [3]. Evolutionary plant breeding (EPB) is an effective approach to increase agrobiodiversity and, thus, contribute to increase yield sustainably, to better adapt to various biotic and abiotic stresses and to bring crop genetic diversity back in farmers' fields and into farmers' hands [4,5].

Research on evolutionary populations and mixtures dates back to 1929 and since the early papers, it was shown that EPB allows the crops to continuously evolve and maintains

its ability to adapt to various biotic and abiotic stresses specially to changing climates [6,7]. EPB is practiced by using evolutionary populations, also called bulk populations, bulks, composites or composites crosses. When it is not possible to establish evolutionary populations, mixtures can be used. The definitions of these terms were given by Wolfe and Ceccarelli [8]. Evolutionary populations are formed by mixing the seed produced from crosses between a number of varieties, while mixtures are obtained by simply mixing the seed of a number of varieties [9]. Mixtures are easier to make in absences of facilities or expertise to make crosses, and landraces are one obvious type of germplasm to use as they are locally adapted and already played an important role for the evolution and conservation of diversity within species [10,11]. Varietal mixture appears as a simple, efficient and cost-effective approach for providing on-farm diversity and to increase the yield gains resulting from a combination of natural and artificial selection [12–14]. The farmers can produce the mixtures by using the already-available varieties in the market, or their traditional landraces. Mixtures can be static or dynamic [8]. Static mixtures are made up by mixing the seed of each of the components at the beginning of each cropping season. They are static because, although such physical seed mixtures are genetically more complex than monocultures and can, therefore, be subjected to natural selection, they do not capture the effects of natural selection occurring in the field. When part of the grain produced from such mixtures is used as seed for the following crop, thus capturing the effects of natural selection, the mixtures are dynamic. Dynamic mixtures tend to become populations [9] over a few generations, even with low levels of out-crossing, as the one occurring in predominantly self-pollinated crops such as rice and beans.

A large body of research spanning over nearly a century shows that evolutionary populations and mixtures tend to become higher yielding than uniform varieties, are more stable over time, improve their disease resistance with time and, in the case of cereals such as wheat and barley, control weeds better than uniform varieties [9]. This was confirmed by a recent meta-analysis [12], which showed that mixtures with more cultivars and those with more functional trait diversity give higher relative yields. However, there are also reports of mixtures of various crops not showing a yield advantage over the components [15] or a better disease protection [16]. Mixtures may perform equal to, or better, or worse than the mean of the components grown in pure stands [17].

Evolutionary populations and mixtures can be used as a tool to create and maintain a high degree of polymorphic variety for accelerating the development of climate resilient and sustainable high performance [18]. Therefore, EPB seems an effective strategy to increase diversity and stability of a crop in a specific environment, can be an effective tool for sustainable food production and minimize climate induced stresses [7,13,19–22].

Except for an experiment on bread wheat [23], we are not aware of other studies using evolutionary populations and mixtures as a source of diversity from which farmers and technical staff can select according to their preferred plant types.

The objectives of this study were (a) to evaluate the agronomic performance and the acceptability by farmers of a number of mixtures of two important crops in the agricultural systems of Nepal, namely rice and bean, and (b) to evaluate the efficiency of superimposing to the natural selection, the selection by farmers, disaggregated by gender, and the selection by technical staff with the perspective of using dynamic mixtures as source material for a participatory breeding program.

## 2. Materials and Methods

The field trials were carried out with rice and beans in farmers' fields in three subsequent years from 2019 to 2021; the trials were conducted in two locations, Jumla and Lamjung in the case of rice and in Jumla in the case of beans. The two crops were chosen based on agro-climatic conditions and farmers' preferences, especially on marginal land of the two sites where the trials were conducted. The germplasm sources for both rice and beans included local landraces, released varieties, breeding lines and landraces obtained both from farmers' field and research stations of Nepal (Table S1).

The germplasm sources used for rice differed in Jumla and Lamjung as the two locations represent agricultural systems at different altitudes. The trial site of Jumla (29°0′04″ N, 80°0′05″ E) is located at 2275 m above sea level (masl) and represents the high-altitude region, whereas Lamjung (28°0′16″ N, 84°0′24″ E) at 850 masl represents the mid-hill region (the location of the two sites is shown in Figure 1).

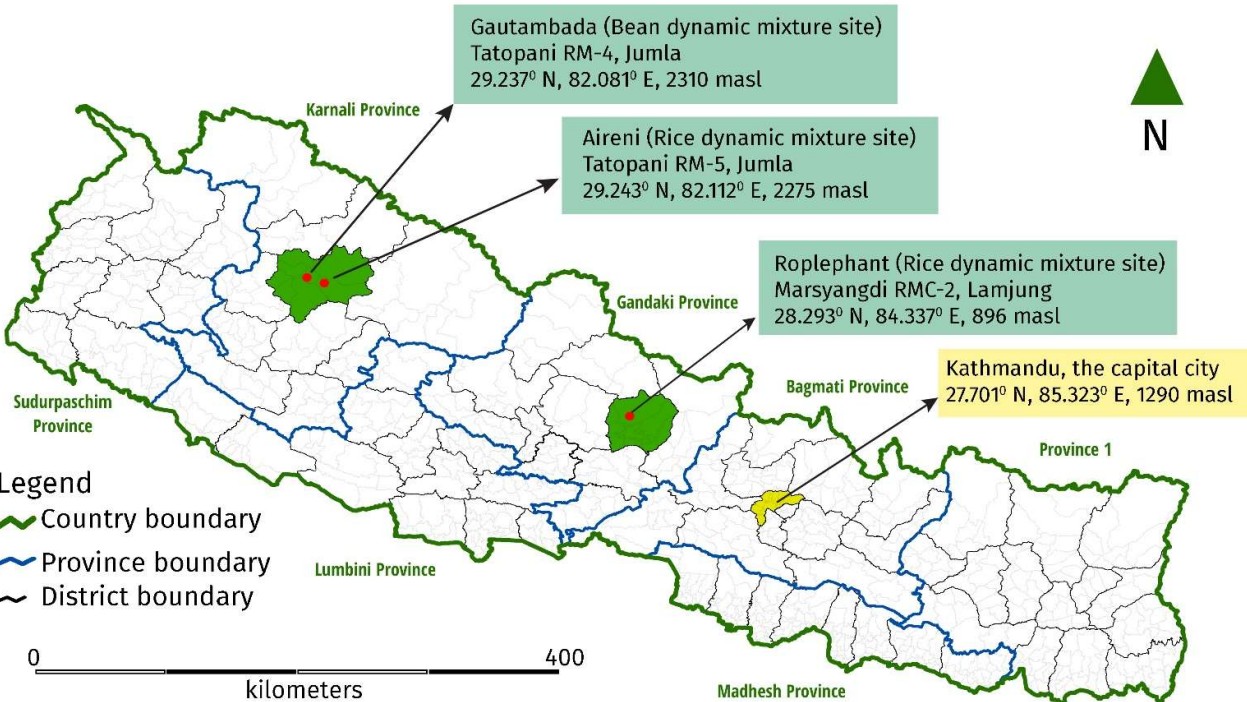

**Figure 1.** Map showing the locations where the trials with dynamic mixtures of rice and beans were conducted for three cropping seasons (2019–2021). The position of Nepal's capital city Kathmandu is shown for reference.

In Jumla, we used 66 rice germplasm sources to develop nine different dynamic mixtures and 48 germplasm sources for constructing similar number of bean dynamic mixtures (Table S1), In Lamjung, we used 56 rice germplasm sources to develop 12 different rice dynamic mixtures (Table S1).

Each mixture was constructed by mixing manually an equal number of seeds of different germplasm for each crop in each location separately based on similar traits (Table 1).

The choice of the source germplasm was discussed with the farmers to incorporate the sources of both abiotic and biotic stress tolerance. To improve the reaction of resultant mixtures to biotic (especially disease) and abiotic (drought) stresses, germplasms known for disease and drought stress tolerance were included during the formation of the mixtures. Similarly, the germplasms sources for each mixture were chosen based on similar leaf and stem characters including grain shape and size of the variety. The disease reaction was assessed based on different reaction capacity against insect pest and diseases, leaf and stem characters including color, size and scent to obtain disease tolerance in each component mixture. The data on disease reaction will be presented in a companion paper.

The germplasm having different contrasting traits were also included comprising different root length, plant height, plant structure, and plant shape and size for ensuring desirable traits in each dynamic mixture. While making each mixture combination, similar maturity time, cooking time or method, and milling quality were considered as important traits. The details of traits considered for the constitution of the mixtures are shown in Table 1.

**Table 1.** Traits considered for the constitution of mixtures of rice and bean.

| Traits | Germplasm Type |
|---|---|
| Canopy area utilization (below and above ground surface area) | Short, medium and long root length<br>Tall, medium and dwarf plant<br>Different shape, size and color (stem and leaf) |
| Insect and disease tolerance | Different reaction to insect pests and diseases<br>Smooth and rough (uneven) stem and leaves<br>Short and medium awn (for rice) |
| Drought tolerance | Deep rooted<br>Erect leaves<br>Tall, medium and dwarf plant height<br>Large leaf but few in number (for rice) |
| Commonalities among traits within varieties | Similar maturity period<br>Easiness in threshing and threshing method<br>Similar cooking time and method<br>Similar milling quality |

The list of entries and the composition of the different mixtures of rice and bean are reported in Table 2. We used a popular local landrace as local check and a popular improved released/registered variety of that domain as improved check.

**Table 2.** List of entries and composition of the mixtures used for the rice and bean trials in Nepal in 2019, 2020 and 2021 (entries in bold were added in 2020 and 2021).

| Location | Crop | Entry Nr. | Composition | Abbreviations [a] |
|---|---|---|---|---|
| Jumla | Rice | 1 | 4 local landraces | Ri_J_Mix1 |
| | | 2 | 37 landraces from similar agro-ecological domains | Ri_J_Mix2 |
| | | 3 | 6 improved cultivars | Ri_J_Mix3 |
| | | 4 | 41 landraces, 6 improved cultivars and 19 advanced breeding lines | Ri_J_Mix4 |
| | | 5 | Jumli Marshi (Local Check) | Jumli Marshi |
| | | 6 | Chandannath-3 (Improved Check) | Chandannath-3 |
| | | 7 | Selected from selection plot by male farmers | Ri_J_M_sel |
| | | 8 | Selected from selection plot by female farmers | Ri_J_F_sel |
| | | 9 | Selected from selection plot by technicians | Ri_J_T_sel |
| Jumla | Bean | 1 | 21 local landraces | Be_J_Mix1 |
| | | 2 | 21 landraces from similar agro-ecological domains | Be_J_Mix2 |
| | | 3 | 6 breeding lines from Jumla | Be_J_Mix3 |
| | | 4 | 42 landraces and 6 breeding lines | Be_J_Mix4 |
| | | 5 | Kalo Male (Local Check) | Kalo Male |
| | | 6 | Trishuli (Improved Check) | Trishuli |
| | | 7 | Selected from selection plot by male farmers | Be_J_M_sel |
| | | 8 | Selected from selection plot by female farmers | Be_J_F_sel |
| | | 9 | Selected from selection plot by technicians | Be_J_T_sel |
| Lamjung | Rice | 1 | 21 local landraces | Ri_La_Mix1 |
| | | 2 | 25 landraces from similar agro-ecological domains | Ri_La_Mix2 |
| | | 3 | 5 improved cultivars | Ri_La_Mix3 |
| | | 4 | 46 landraces, 5 improved cultivars and 5 advanced breeding lines | Ri_La_Mix4 |
| | | 5 | Gaure (Local check) | Gaure |
| | | 6 | Khumal 4 (Improved check) | Khumal 4 |
| | | 7 | Selected early from selection plot by male farmers | Ri_La_EM_sel |
| | | 8 | Selected early from selection plot by female farmers | Ri_La_EF_sel |
| | | 9 | Selected early from selection plot by technicians | Ri_La_ET_sel |
| | | 10 | Selected late from selection plot by male farmers | Ri_La_LM_sel |
| | | 11 | Selected late from selection plot by female farmers | Ri_La_LF_sel |
| | | 12 | Selected late from selection plot by technicians | Ri_La_LT_sel |

[a] Abbreviation: Ri = Rice, Be = Bean, J = Jumla, La = Lamjung, Mix = mixture, M = male, F = female, T = technician, sel = selection, E = early, L = late.

The randomized complete block design (RCBD) with three replications was used to lay out the experiment in both rice and bean. Randomization was different in each year and each location and was generated by DiGGer [24], a program that generates efficient experimental designs for non-factorial experiments with plots arranged in rows and columns [25,26]. In 2019, the plot size was 20 m² (5 m × 4 m) for rice both in Lamjung and Jumla, and 30 m² (6 m × 5 m) for bean with 0.5 m alley between plots. Due to land availability, in 2020 and 2021, plot size in Jumla was reduced to 12 m² (4 m × 3 m) in the case of rice and to 18 m² (6 m × 3 m) for bean with 0.5 m alleys. In the case of Lamjung, plot size was 10 m² (5 m × 2 m) in 2020 and 9 m² (3 m × 3 m) in 2021.

The traits measured in the rice trials were:

1. Plant height (PH in cm);
2. Number of tillers (NT);
3. Panicle length (PL in cm);
4. Number of panicles per plant (NP);
5. Days to 50% flowering (DF in days);
6. Days to 80% maturity (DM in days);
7. Thousand grain weight (TGW in g);
8. Grain yield (GY in kg ha$^{-1}$).

Traits 1 to 4 were measured on five to fifteen random plants per plot, while traits 5 to 8 were measured on a plot basis.

The traits measured in the bean trials were:

1. Plant height (PH in cm);
2. Number of pods per plant (PodP);
3. Pod length (PodL in cm);
4. Days to 50% flowering (DF in days);
5. Days to 80% maturity (DM in days);
6. Thousand grain weight (TGW in g);
7. Grain yield (GY in kg ha$^{-1}$).

Traits 1 to 3 were measured on five to fifteen random plants per plot, while traits 5 to 8 were measured on a plot basis.

In the case of rice, plant height (PH), number of tillers per hill (NT), panicle length (PL) and number of panicles per plant (NP) were measured at the time of maturity. DF was measured when 50% of the plants in a plot flowered and recorded as plot basis. DM was recorded in rice when 80% of the grains on the panicle were fully ripened and grain became hard when chewed. In the case of beans, we measured DM when 80% of the pods turned into brown and yellow color and the leaves of the plants also turned into yellow color. In bean, PodP and PodL were measured at the time of maturity. Grain yield and TGW were measured in grams on a plot basis. Grain yield was adjusted for moisture at 13% and 10%, for rice and bean, respectively, and was then converted into kilograms per hectare (kg ha$^{-1}$) in both crops. The adjustment for moisture content was performed following Badu-Apraku et al. [27] as:

$$GYadj = \frac{100 - AMC(\%)}{100 - SMC(\%)} * 100$$

where *AMC* is the actual measured moisture content and *SMC* = standard moisture content (13% for rice and 10% for beans)

The agronomic traits and their rating scale were taken according to the protocol of Bioversity International, IRRI and WARDA [28] for rice and according to the descriptors of the International Board for Plant Genetic Resources [29] for bean.

Additionally, we also established selection plots in 2019 next to the trials by using the mixture of landraces, improved varieties and breeding lines of each crop, namely entry 4 (Table 2). The size of the selection plot was 40 m² for rice and 60 m² for bean. Both for Jumla and Lamjung, the selection plot of rice and beans was divided into four equal

sub-plots for imposing selection by male farmers, female farmers and technicians. Each of the three subplots were used for selection by male, female and technicians separately with a mixture of all the components and the fourth sub-plot was simply ignored as this was not assigned to any specific group. The technicians were from the Agriculture Service Centers and extension services outside breeding institutions. Additionally, the technical staff/field technicians of LI-BIRD were included. Each group selected 100 plants in the case of rice and 50 plants in the case of beans with desirable traits. Individual plant selection was carried out once during flowering and once before harvest stage using different marking threads for the identity of selected plants in rice and bean, respectively. The selected plants were harvested and their seed bulked.

Therefore, in Jumla, for both crops, the 2019 trials included four mixtures and two checks; in 2020, three new entries were added, which were the results of the selection carried out by male farmers, female farmers and technicians based on the preferred traits. These three additional entries were also evaluated in 2020 and 2021 trials. Therefore, for both years, the trials included nine entries each in Jumla for rice and beans.

In Lamjung, the selection protocol was slightly different because the three groups of selectors (male farmers, female farmers and technicians) performed the selection on their respective sub-plot two consecutive times based on physiological maturity attained by an individual plant. The improved check (Khumal 4) of Lamjung represents an early variety; hence, those plants which reached the physiological maturity around the similar period of Khumal 4 (Entry 6) (Table 2) represent early selected dynamic mixtures from the selection plot. Similarly, Gaure (local check) represents the late variety in this location. Hence, we produced late selected dynamic mixtures by selecting those plants which attained physiological maturity period around Gaure (Entry 5) (Table 2). We used different colored threads to recognize early and late selected dynamic mixtures within the same sub-plot while harvesting. This resulted in six additional entries which were added in 2020 and 2021 trials, which, therefore, included twelve entries (Table 2).

The preference ranking was carried out by five male farmers, five female farmers and five technicians twice, during flowering and before harvest. The farmers were mainly engaged from community seed banks for the selection of promising entries in the trials. The selection criteria were set based on key preferred traits for each rice and bean entry in both the sites. The major selection criteria in both rice and bean included high yield, long panicles or pods, high tillering capacity, disease resistance/tolerance, early-to-late maturity depending on the cropping season and location, high number of pods per plant and grains per panicle, high pod yield and uniform maturity. The farmers were asked to express their preference based on the above-mentioned traits. A small container capped with paper cover with a hole at the center was placed in each plot for dropping votes. The farmers/technicians were requested to select the three most preferred entries through a voting process. However, during voting in the field, each farmer was requested to cast the vote one by one without discussing and coming to an agreement with each other. The farmers cast their votes by using the grains provided to them. Each male farmer, each female farmer and each technician were provided with six grains of distinguishable crops for voting. Each of them dropped three grains for the best preferred variety, two grains for the second preferred variety and one grain for the third preferred variety. Finally, the votes deposited in the individual variety's container were counted separately for male farmer, female farmer and technician, and were then ranked based on the total votes collected in each variety's container. The most promising entries were identified and recorded in tabular form for their analyses.

During selection, farmers and technicians assembled the plants with the most preferred traits in each dynamic mixture based on their local adaptation, community needs and market value. Similarly, the negative selection was carried out mostly to remove the effect of undesirable plants within each inter-varietal mixture. The selected plants were identified using different colored threads in the field according to the different selection groups. Those selections were bulk harvested separately. At the end of each cropping season,

bulk harvested seeds from each entry were taken separately as the source population for next season.

The agronomic management (weeding, fertilizer and irrigation) was carried out as per the farmers' practice of the respective growing areas.

The data were submitted to a combined analysis of variance by location, and in the case of Jumla, by crop using the ANOVA command for unbalanced designs in GenStat 22nd edition [30], using a random model. In the case of data collected on individual plants within each plot, we used a model in which the Entry trait *Yijz* was a function of the grand mean μ, of the Entry (*E*) effect of the *i* entry, of the Year effect (*Y*) of the *j* year, of the E × Y(*EY*) interaction effect, of the plants within entries (P$_E$) effect and of the residual *e*:

$$Yijz = \mu + Ei + Yj + EYij + P_Ei + eijz$$

In the case of data collected on a plot basis, the model did not include the effect of the plants within entries.

The relationships between traits were analyzed by the GGE biplot package [31] available in R [25]. In the GGE biplot, the use of standardized data made it possible to analyze data measured on different scales. We only used one feature of the GGE biplot, namely the relations between traits. In this feature, the angle between the vectors connecting the labels corresponding to each trait and the origin were proportional to the correlation between the corresponding traits: an angle < 90° indicates a positive correlation, an angle > 90° indicates a negative correlation and angle around 90° indicates independence [31]. This applies as long as the data are sufficiently approximated by the biplot. The information about the relationships between traits generated by the biplots was validated by calculating the Pearson correlation coefficients between the means of all the traits.

## 3. Results

The results are presented by crop and by location.

Before presenting the results, we should remember that the trials were unbalanced, as the selections by farmers and technical staff were added in the second year and this could have affected the comparisons between the selections and the entries (mixtures and controls) that were present in the first years. However, this should not be the case because, using grain yield as an indicator, the three years were either rather similar, as in the case of rice in Lamjung (Figure 2), or the year with the lowest yield was 2020, as in the case of Jumla for both rice and bean. Therefore, the entries evaluated only in 2020 and 2021 should have been affected more than those evaluated for three years.

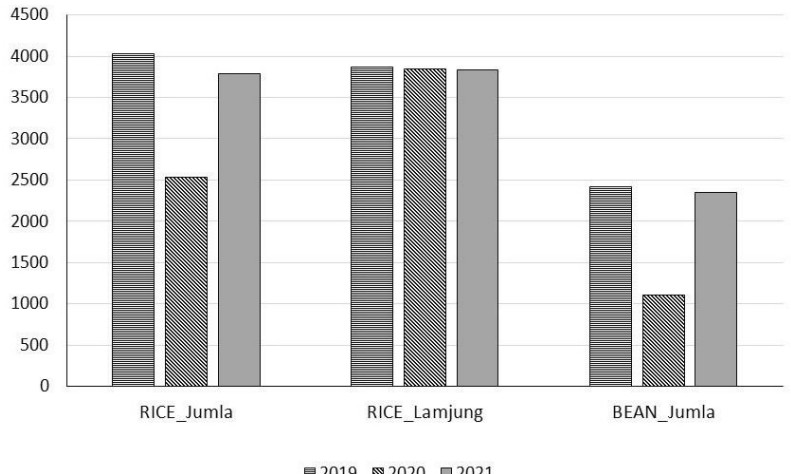

**Figure 2.** Grain yield grand means (kg ha$^{-1}$) of three trials conducted in two locations (Jumla and Lamjung) on rice and one location (Jumla) on beans for three years.

### 3.1. Rice in Jumla

There was always a large year effect on both phenological (DF and DM) and agronomic traits. Despite the presence of a significant entries x years interaction for most traits with the exception of PH, NT TGW and GY, the difference between entries was also always highly significant (Table 3).

**Table 3.** ANOVA of days to flowering (DF), days to maturity (DM), plant height (PH), number of tillers (NT), number of panicles (NP), panicle length (PL), thousand grain weight (TGW) and grain yield (GY) measured in seven rice mixtures, one local landrace and one improved variety evaluated for three years in Jumla, Nepal (entries 7 to 9 were evaluated only during the last two years).

| Source of Variation | DF | DM | PH | NT | NP | PL | TGW | GY [a] |
|---|---|---|---|---|---|---|---|---|
| Entries (E) | 116.7 *** | 110.3 *** | 1041.7 *** | 78.0 *** | 65.9 *** | 87.3 *** | 16.1 *** | 39.9 ** |
| Year (Y) | 31,822.7 *** | 44,043.8 *** | 2069.4 *** | 1013.1 *** | 784.9 *** | 460.4 *** | 356.3 *** | 136.6 *** |
| E x Y | 9.2 ** | 11.6 *** | 131.4 | 8.6 | 9.3 | 11.4 * | 3.8 | 15.1 |
| Plants w. E ($P_E$) | - | | 166.0 | 14.3 * | 14.0 ** | 6.5 | | |
| Residual | 3.29 | 0.44 | 148.30 | 10.0 | 9.8 | 5.68 | 3.8 | 12.7 |

[a] Original values $\times 10^{-4}$; * = $p < 0.05$; ** = $p < 0.01$; *** = $p < 0.001$.

Most entries were later in flowering and maturity than the local check Jumli Marshi (Table 4). The male selections were the latest maturing entries and differed significantly from both the female selections and even more from the technician selections. All the entries were taller than the local check except RI_J_Mix3. However, the mixture of four local landraces (Ri_J_Mix1), the mixture of 37 landraces from similar agro-ecological domains (Ri_J_Mix2), and the mixture of six improved cultivars (Ri_J_Mix3) were different from the improved check. The three selected mixtures (Ri_J_M_sel, Ri_J_F_sel and Ri_J_T_sel) were among the tallest entries not significantly different from the improved check (Chandannath-3). The three selected mixtures had a significantly higher number of tillers and of panicles compared to local and improved check. The local check had the shortest panicles not significantly different from Ri_J_Mix1, composed of local landraces. The TGW of the selections carried out by male farmers and female farmers were the largest and were significantly higher than the improved check but not significantly different from the local check. Ri_J_Mix1 and Ri_J_Mix3 yielded significantly more than the local check and were not significantly different from the improved check. Only the yield of the male selections matched the yield of the improved check and of highest yielding mixtures.

**Table 4.** Means [b] across years of days to flowering (DF in days), days to maturity (DM in days), plant height (PH in cm), number of tillers (NT), number of panicles (NP), panicle length (PL in cm), thousand grain weight (TGW in g) and grain yield (GY in kg ha$^{-1}$) measured in seven rice mixtures, one local landrace (Jumli Marshi) and one improved variety (Chandannath-3) evaluated for three years in Jumla, Nepal (entries 7 to 9 were evaluated in 2020 and 2021).

| Entry Name [a] | DF | DM | PH | NT | NP | PL | TGW | GY |
|---|---|---|---|---|---|---|---|---|
| Ri_J_Mix1 | 100 cd | 148.4 f | 91.8 de | 6.5 b | 6.4 b | 18.9 de | 30.8 cd | 3518 a |
| Ri_J_Mix2 | 103 b | 149.8 e | 92.0 cde | 6.7 b | 6.8 b | 20.2 bc | 31.6 abcd | 3394 ab |
| Ri_J_Mix3 | 101.7 bc | 151.8 d | 89.5 e | 7.1 b | 7.1 b | 19.8 cd | 29.8 d | 3495 a |
| Ri_J_Mix4 | 100.6 cd | 150.4 e | 94.8 bcd | 6.8 b | 6.6 b | 19.7 cd | 30.9 bcd | 3306 abc |
| Jumli Marshi | 98.7 d | 148.1 f | 91.2 de | 6.6 b | 6.6 b | 18.1 e | 33.0 ab | 3082 bc |
| Chandannath-3 | 103.3 b | 151.5 d | 98.6 ab | 6.8 b | 6.7 b | 20.8 ab | 30.7 d | 3140 abc |
| Ri_J_M_sel | 106.1 a | 155.0 a | 99.4 a | 9.4 a | 9.1 a | 21.6 a | 31.6 abcd | 3230 abc |
| Ri_J_F_sel | 106.4 a | 153.5 c | 96.1 abc | 8.6 a | 8.3 a | 20.8 ab | 32.8 abc | 2925 c |
| Ri_J_T_sel | 105.9 a | 154.2 b | 96.6 ab | 8.5 a | 8.4 a | 20.6 bc | 33.6 a | 2949 c |

[a] Abbreviations as in Table 2; [b] Means with a letter in common are not significantly different ($p < 0.05$).

There were significant differences between the preferences for the nine entries at the flowering stage where Ri_J_Mix1 (the mixture of four local landraces) was the most preferred (Table 5). At maturity, we found significant differences only for the evaluation given by technicians and the same mixture (Ri_J_Mix1) was the second most preferred after Jumli Marshi. At the same stage, we found non-significant differences for the evaluation given by female and male farmers. Ri_J_Mix1 was the most preferred variety by female at maturity; however, we found that female selections (Ri_J_F_sel) were the most preferred by males, followed by Ri_J_M_sel and Ri_J_Mix3, respectively. In the case of female and male evaluation, particularly at flowering, Ri_J_Mix1 was not significantly different from Jumli Marshi, and in the case of males, it was not significantly different from other two mixtures, namely Ri_J_Mix2 and Ri_J_Mix4. The mean evaluations given by females, males and technical staff at flowering stage (MeanFL) suggest that Ri_J_Mix1 received a significantly better score value followed by Jumli Marshi, Ri_J_Mix2 and Ri_J_Mix4, respectively. Moreover, for MeanFL, the selections did not differ from each other, and especially from Chandannath-3; however, we observed differences from Ri_J_Mix1 and Jumli Marshi. There were no significant differences between the preferences given at maturity.

**Table 5.** Farmers' and technicians' preferences [b] on seven mixtures, one local landrace and one improved variety evaluated for three years in Jumla, Nepal (entries 7 to 9 were evaluated only during the last two years).

| Entry Name [a] | FFL | MFL | TFL | FH | MH | TH | MeanFL | MeanH |
|---|---|---|---|---|---|---|---|---|
| Ri_J_Mix1 | 11.6 a | 9.5 a | 7.9 a | 5.3 | 4.1 | 6.1 a | 9.9 a | 3.8 |
| Ri_J_Mix2 | 4.4 bc | 4.9 abc | 4.2 b | 2.6 | 3.7 | 2.9 ab | 4.7 bc | 2.5 |
| Ri_J_Mix3 | 1.9 c | 3.7 bc | 0.8 b | 3.5 | 4.4 | 1.4 b | 2.2 c | 3.6 |
| Ri_J_Mix4 | 5.2 bc | 5.6 abc | 2.1 b | 2.2 | 3.4 | 2.4 ab | 4.1 bc | 2.9 |
| Jumli Marshi | 7.6 ab | 7.9 ab | 2.1 b | 3.5 | 2.0 | 6.9 a | 6.7 b | 3.1 |
| Chandannath-3 | 3.0 c | 2.9 c | 0.8 b | 4.7 | 2.3 | 4.6 ab | 2.4 c | 3.4 |
| Ri_J_M_sel | 4.0 bc | 4.3 bc | 0.0 c | 3.3 | 4.5 | 3.0 ab | 3.1 c | 4.1 |
| Ri_J_F_sel | 1.5 c | 3.7 bc | 3.3 b | 3.7 | 6.5 | 4.0 ab | 2.3 c | 5.1 |
| Ri_J_T_sel | 5.3 bc | 1.8 c | 7.0 a | 3.0 | 0.8 | 1.3 b | 3.7 bc | 1.8 |
| $P_{F(Entries)}$ | 0.001 | 0.015 | 0.0006 | 0.42 | 0.198 | 0.032 | <0.0001 | 0.747 |

[a] Abbreviations as in Table 2; [b] Means with a letter in common are not significantly different ($p < 0.05$); FFL = females' score at flowering; MFL = males' score at flowering; TFL = technical staff score at flowering; FH = females' score at harvesting; MH = males' score at harvesting; TM = technical staff score at harvesting; MeanFL = average score at flowering; MeanH = average score at harvesting.

The relationships between agronomic traits and preferences (Figure 3) show a positive correlation between panicle length (PL), days to flowering (DF), days to maturity (DM), plant height (PH), number of tillers (NT) and number of panicles (NP). The correlation coefficients (Table S2) ranged from 0.997 ($p < 0.01$) between NP and NT to 0.697 ($p < 0.05$) between PH and DM. Only the correlation coefficient between NT and PH (r = 0.652 $p = 0.076$) was not significant. The selections made by the technical staff (T_sel) were more closely associated with thousand grain weight (TGW). The evaluations made at flowering by females (FFL), males (MFL), and by the technical staff at maturity (TH) as well as the average score at flowering (MeanFL) were negative correlated with PL, DF, DM, PH, NT, NP and PL, although only the correlations between MFL and DF (r = −0.762 $p < 0.05$), MFL and DM (r = −0.796 $p < 0.05$) and MFL and PL (r = − 0.770 $p < 0.05$) were significant. They tend to indicate a preference for Jumli Marshi and for the mixture of four local varieties (Ri_J_Mix1). Although at maturity, the differences were only significant in the case of the evaluation by the technical staff (Table 5). There was a tendency of positive correlation for both the male evaluation and the average to be associated with improved germplasm (Chandannath-3 and Ri_J_Mix 3). The selections carried out by the technical staff were lower yielding as opposed to the selections by males and females (Figure 3).

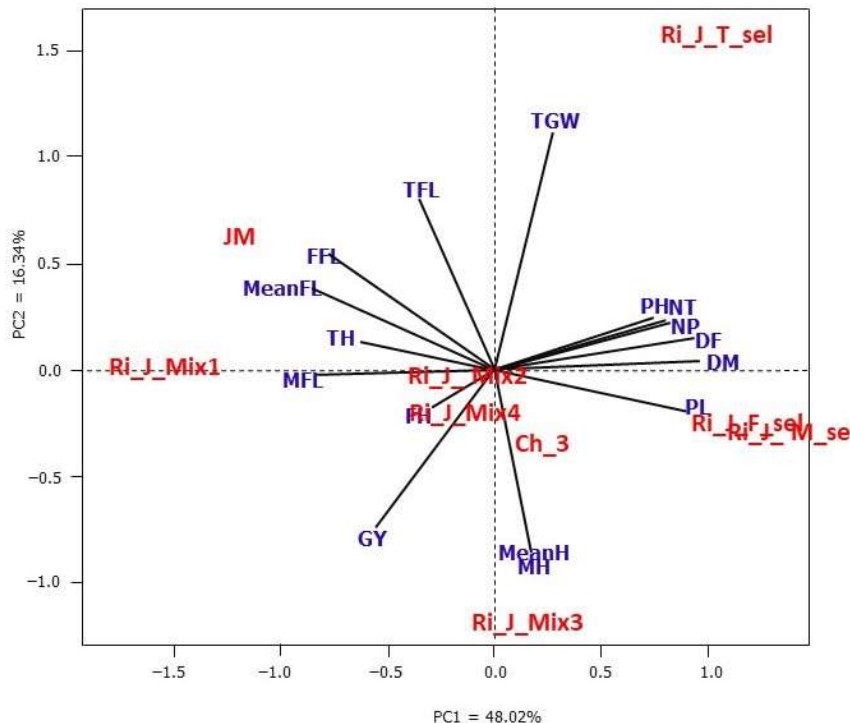

**Figure 3.** Biplot showing the relationships between traits, including agronomic evaluation, on seven mixtures, one local landrace and one improved variety of rice grown in Jumla, Nepal, for three cropping seasons, 2019–2021 (Ri_J_F_sel, Ri_J_M_sel and Ri_J_T_sel were grown only during 2020 and 2021 cropping season).

### 3.2. Rice in Lumjung

Contrary to what was observed in Jumla, in Lamjung, the year effect was significant only on days to maturity, plant height, number of tillers, number of panicles and thousand grain weight. There was a significant entries x years interaction for most traits with the exception of NT, NP and GY. Nevertheless, the difference between entries was also significant for most traits with the exception of PL and TGW (Table 6).

**Table 6.** ANOVA of days to flowering (DF), days to maturity (DM), plant height (PH), number of tillers (NT), number of panicles (NP), panicle length (PL), thousand grain weight (TGW) and grain yield (GY) measured in ten rice mixtures, one local landrace and one improved variety evaluated for three years in Lamjung, Nepal (entries 7 to 12 were evaluated only during the last two years).

| Source of Variation | DF | DM | PH | NT | NP | PL | TGW | GY [a] |
|---|---|---|---|---|---|---|---|---|
| Entries (E) | 679 *** | 715.8 *** | 17,559.1 *** | 2.2 ** | 2.5 *** | 71.9 | 8.9 | 145.3 *** |
| Year (Y) | 33.3 | 38.5 * | 4095.4 * | 2.3 * | 3.2 * | 38.9 | 54.7 ** | 3.4 |
| E x Y | 35.9 *** | 9.6 *** | 873.0 *** | 0.5 | 0.8 | 35.1 *** | 7.8 ** | 24.7 |
| Plants w. E | - | - | 338.0 | - | - | 7.7 | - | - |
| Residual | 2.00 | 0.43 | 241.7 | 0.70 | 0.65 | 7.7 | 2.6 | 34.6 |

[a] Original values $\times 10^{-4}$; * = $p < 0.05$; ** = $p < 0.01$; *** = $p < 0.001$.

　　　　All the entries were significantly earlier in flowering and maturity than the local check Gaure (Table 7). The early selections were significantly different from Khumal 4 in terms of flowering time. Male and female selections were significantly earlier than Khumal 4, while the technicians' selections were significantly later than Khumal 4, indicating that males and females preferred the varieties even earlier than the improved check. The mixture of improved varieties (Ri_La_Mix3) had the same flowering time as the improved check and was marginally, but not significantly, later in maturity.

**Table 7.** Means [b] across years of days to flowering (DF in days), days to maturity (DM), plant height (PH in cm), number of tillers (NT), number of panicles/plant (NP), panicle length (PL in cm), thousand grain weight (TGW in g) and grain yield (GY in kg ha$^{-1}$) measured in ten rice mixtures, one local landrace and one improved variety evaluated for three years in Lamjung, Nepal (entries 7 to 12 were evaluated only during the last two years).

| Entry Name [a] | DF | DM | PH | NT | NP | PL | TGW | GY |
|---|---|---|---|---|---|---|---|---|
| Ri_La_Mix1 | 116.4 e | 167.7 cd | 136.0 d | 8.0 a | 7.5 a | 22.8 | 26.7 | 4214 ab |
| Ri_La_Mix2 | 126.6 c | 167.2 d | 141.7 c | 7.4 ab | 7.1 ab | 23.2 | 28.7 | 4039 abc |
| Ri_La_Mix3 | 108.3 g | 149.3 ef | 107.9 g | 6.6 bc | 5.5 d | 21.7 | 26.1 | 3424 cde |
| Ri_La_Mix4 | 116.9 e | 167.7 cd | 133.4 d | 7.4 ab | 7.0 abc | 22.2 | 27.6 | 4053 abc |
| Gaure | 134.7 a | 172.7 a | 143.7 c | 6.6 bc | 6.3 bcd | 24.0 | 26.8 | 4065 abc |
| Khumal 4 | 108.0 g | 148.3 g | 114.8 f | 7.9 a | 7.6 a | 21.9 | 24.4 | 3630 bcd |
| Ri_La_EM_sel | 105.6 h | 148.2 g | 120.6 e | 6.6 bc | 5.8 d | 21.4 | 27.4 | 3208 de |
| Ri_La_EF_sel | 104.1 h | 149.8 e | 119.9 ef | 6.7 bc | 6.3 bcd | 21.4 | 25.9 | 2908 e |
| Ri_La_ET_sel | 111.1 f | 148.8 fg | 123.2 e | 7.2 abc | 6.4 bcd | 22.1 | 26.7 | 3114 de |
| Ri_La_LM_sel | 128.8 b | 168.6 b | 152.0 a | 6.4 bc | 6.2 bcd | 24.6 | 28.4 | 4478 a |
| Ri_La_LF_sel | 129.4 b | 168.2 bc | 149.2 ab | 6.3 c | 6.0 cd | 24.1 | 27.9 | 4348 a |
| Ri_La_LT_sel | 121.1 d | 168.0 bc | 144.3 bc | 6.0 c | 5.8 d | 23.8 | 27.7 | 4260 ab |
| P$_{F(Entries)}$ | 0.001 | 0.001 | 0.001 | 0.003 | 0.001 | 0.093 | .397 | 0.005 |

[a] Abbreviations as in Table 2; [b] Means with a letter in common are not significantly different (*p* < 0.05).

　　　　The local landrace was the tallest entry after the late selected entries by males, females and technical staff (LM_Sel_Ri, LF_Sel_Ri and LT_Sel_Ri). We found that the mixture of five improved varieties (Ri_La_Mix3) was the shortest and significantly shorter than Khumal 4. However, Khumal 4 did not differ in height from the early female selections (Ri_La_EF_sel). The tallest entries were the late selections, regardless of who made the selection. Khumal 4 had the higher number of tillers and of panicles together with three mixtures (Ri_La_Mix1, Ri_La_Mix2 and Ri_La_Mix4) and, only related to number of tillers, with the early selections by the technicians. The highest grain yields were obtained by the late selections by males, females and technical staff, respectively (Table 7). These male and female late selections yielded significantly more than Khumal 4 but did not differ significantly from the local check Gaure and from three mixtures (Ri_La_Mix1, Ri_La_Mix2 and Ri_La_Mix4). The lowest yielding entries were the early selections by females, technicians and males, and a mixture of five improved varieties (Ri_La_Mix3).

　　　　Preferences differed significantly with the only exception of the male's preference at flowering (Table 8). This was the only case in which the entry x years interaction was significant (*p* = 0.047), thus masking the differences between entries. The most preferred entries, both at flowering and before harvesting, were the Ri_La_Mix1 (the mixture of 21 local landraces) and the Ri_La_Mix4 (the most complex mixture). The local check Gaure was among the most preferred before harvesting, while the improved check Khumal 4 was among the most preferred by the female farmers at flowering. The early selections, regardless of who made them, were among the least preferred both at flowering and before harvesting. On the contrary, the late selections, and particularly those made by the female farmers, were among the most preferred especially before harvesting.

**Table 8.** Farmers' and technicians' preferences [b] on ten rice mixtures, one local landrace (Gaure) and one improved variety Khumal 4 evaluated for three years in Lamjung, Nepal (entries 7 to 12 were evaluated only during the last two years).

| Entry Name [a] | FFL | MFL | TFL | FH | MH | TH | MeanFL | MeanH |
|---|---|---|---|---|---|---|---|---|
| Ri_La_Mix1 | 5.64 a | 4.75 | 6.21 a | 4.11 abc | 3.95 abc | 6.03 a | 5.53 a | 4.70 ab |
| Ri_La_Mix2 | 1.90 b | 3.28 | 0.68 f | 1.57 cde | 1.31 de | 0.90 de | 1.95 b | 1.26 ef |
| Ri_La_Mix3 | 0.86 b | 1.53 | 1.96 cdef | 1.02 e | 0.84 de | 1.31 de | 1.45 b | 1.05 ef |
| Ri_La_Mix4 | 5.73 a | 3.33 | 5.50 a | 4.82 ab | 4.91 ab | 5.34 a | 4.86 a | 5.02 a |
| Gaure | 2.39 b | 1.92 | 2.84 c | 5.66 a | 5.08 ab | 4.96 a | 2.38 b | 5.24 a |
| Khumal 4 | 5.20 a | 4.35 | 4.62 b | 2.78 bcde | 2.28 cde | 2.97 bc | 4.72 a | 2.67 cde |
| Ri_La_EM_sel | 1.52 b | 3.74 | 1.16 def | 1.36 de | 1.95 cde | 1.14 de | 2.14 b | 1.48 def |
| Ri_La_EF_sel | 1.76 b | 1.97 | 1.35 cdef | 1.13 de | 0.54 e | 0.94 de | 1.69 b | 0.87 f |
| Ri_La_ET_sel | 1.56 b | 2.14 | 2.57 cd | 0.55 e | 0.75 de | 2.37 bcd | 2.09 b | 1.22 ef |
| Ri_La_LM_sel | 1.74 b | 1.54 | 1.96 cdef | 2.56 bcde | 3.18 bcd | 3.36 bc | 1.75 b | 3.03 bcd |
| Ri_La_LF_sel | 2.36 b | 3.18 | 2.37 cde | 3.77 abc | 5.74 a | 2.16 cde | 2.64 b | 3.89 abc |
| Ri_La_LT_sel | 1.52 b | 1.15 | 0.95 ef | 3.33 abcd | 2.54 cde | 0.77 e | 1.21 b | 2.21 cdef |
| $P_{F(Entries)}$ | 0.0003 | 0.174 | <0.00001 | 0.0002 | 0.00006 | <0.00001 | <0.00001 | <0.00001 |

[a] Abbreviations as in Table 2; [b] Means with a letter in common are not significantly different ($p < 0.05$); FFL = females' score at flowering; MFL = males' score at flowering; TFL = technical staff score at flowering; FH = females' score at harvesting; MH = males' score at harvesting; TM = technical staff score at harvesting; MeanFL = average score at flowering; MeanH = average score at harvesting.

Plant height, (PH), panicle length (PL), days to flowering (DF), days to maturity (DM), grain yield (GY) and thousand grain weight (TGW) were positively correlated (Figure 4). These correlation coefficients ranged from 0.936 ($p < 0.01$) to 0.593 ($p < 0.05$) and only the correlation coefficient between GY and TGW (r = 0.554, $p = 0.097$) (Table S3) was not significant. The high expression of these traits, namely tall plants, long panicles and late flowering and maturity expressed in the local landrace Gaure, received a high score at maturity by male (MH) and females (FH) and as average score at maturity (MeanH). The late female selections (Ri_La_LF_sel) were for this expression of these traits. On the contrary, and as indicated by their position, the early selections by females, males and technicians (Ri_La_EF_sel, Ri_La_EM_sel and Ri_La_ET_sel) were for the opposite expression of the same traits (short plants with short panicles, early flowering and maturity and low yield as expressed in the mixture of improved varieties (Ri_La_Mix3)). Furthermore, the early selections and the late selections were clearly separated.

Selection at flowering (FFL, MFL, TFL and MeanFL) was for a high number of tillers (NT) and of panicles (NP), which were expressed in the mixture of 21 local landraces (Ri_La_Mix1) and in Khumal 4. High number of tillers (NT) and of panicles (NP) were negatively associated with thousand grain weight (TGW) but the correlation coefficients (r = −0.395 and r = −0.266) were not significant (Table S3).

*3.3. Bean in Jumla*

In the bean trials in Jumla, entries x years interactions were always large and highly significant (Table 9), making differences between entries hardly detectable except for plant height (PH), pod length (PodL) and thousand grain weight (TGW). The differences between years were also very large and often highly significant.

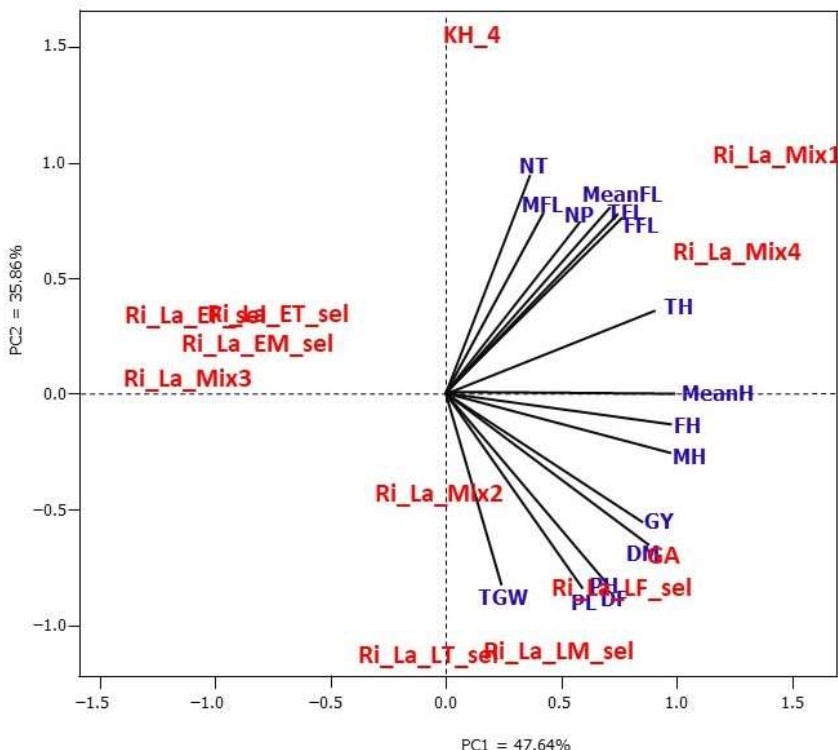

**Figure 4.** Biplot showing the relationships between traits, including agronomic evaluation on ten mixtures and two checks of rice grown in Lamjung, Nepal, for three cropping seasons. Ri_La_EF_sel, Ri_La_EM_sel, Ri_La_ET_sel, Ri_La_LF_sel, Ri_La_LM_sel and Ri_La_ LT_sel were grown only during 2020 and 2021 cropping seasons.

**Table 9.** ANOVA of days to flowering (DF), days to maturity (DM), plant height (PH), number of pods per plant (PodP), pod length (PodL), 1000 grain weight (TGW) and grain yield (GY) measured in nine bean mixtures evaluated for three years in Jumla, Nepal (entries 7 to 9 were evaluated only during the last two years).

| Source of Variation | DF | DM | PH | PodP | PodL | TGW | GY [a] |
|---|---|---|---|---|---|---|---|
| Entries (E) | 76.8 | 10.9 | 22,630.9 * | 299.9 | 406.9 *** | 4810.4 * | 770.8 |
| Year (Y) | 446.4 *** | 145.8 ** | 39,859.8 * | 2965.9 ** | 179.3 *** | 11,056.2 ** | 13,217.8 *** |
| E x Y | 34.0 *** | 15.2 *** | 7245.2 *** | 277.4 *** | 8.2 ** | 1703.3 *** | 466.4 * |
| Plants w. E | - | - | 363.1 | 12.7 | 3.2 | | |
| Residual | 0.3 | 1.4 | 466.6 | 12.6 | 3.7 | 248.6 | 201.2 |

[a] Original values $\times 10^{-3}$; * = $p < 0.05$; ** = $p < 0.01$; *** = $p < 0.001$.

Trishuli, the improved check, was significantly taller than all other entries and was also characterized by longer pods and lighter seed (Table 10). The mixtures, with the exception of Be_J_Mix3 (the mixture of breeding lines), were significantly shorter than the local landrace Kalo Male used as check, like the selections made by female and male farmers and technicians (Be_J_M_sel, Be_J_F_sel and Be_J_T _sel). The local landrace (Kalo Male) had the largest grains together with the mixture of breeding lines (Be_J_Mix3). All the other entries had a thousand grain weight (TGW) that was intermediate between Kalo Male and Trishuli. Although differences were not significant, we noted a tendency of the mixture of landraces produced from similar agro-ecological domains of Nepal (Be_J_Mix2) to have a higher grain yield followed by the male selected mixture and the mixture of local landraces of the sites, respectively.

**Table 10.** Means [b] across years of days to flowering (DF in days), days to maturity (DM), plant height (PH in cm), number of tillers (NT), number of pods per plant (PodP), pod length (PodL in cm), 1000 grain weight (TGW in g) and grain yield (GY in kg ha$^{-1}$) measured in nine bean mixtures evaluated for three years in Jumla, Nepal (entries 7 to 9 were evaluated only during the last two years).

| Entry Name [a] | DF | DM | PH | PodP | PodL | TGW | GY |
|---|---|---|---|---|---|---|---|
| Be_J_Mix1 | 52.8 | 92.9 | 47.7 c | 7.56 | 10.4 b | 247.8 bc | 1972 |
| Be_J_Mix2 | 52.6 | 92.7 | 45.6 c | 8.83 | 10.2 b | 226.5 d | 2037 |
| Be_J_Mix3 | 50.3 | 92.9 | 58.0 b | 8.04 | 9.9 bc | 272.1 a | 1899 |
| Be_J_Mix4 | 52.3 | 93.3 | 48.9 c | 8.18 | 10.0 bc | 245.8 bc | 1969 |
| Kalo Male | 47.4 | 92.6 | 57.5 b | 8.32 | 9.1 d | 279.5 a | 1874 |
| Trishuli | 56.8 | 95.1 | 84.0 a | 10.15 | 15.8 a | 197.1 e | 955 |
| Be_J_M_sel | 49.5 | 93.7 | 49.9 c | 7.01 | 9.8 bc | 238.3 bcd | 1994 |
| Be_J_F_sel | 47.8 | 93.9 | 45.6 c | 6.39 | 9.5 cd | 254.4 b | 1674 |
| Be_J_T_sel | 48.0 | 93.5 | 46.6 c | 6.48 | 10.1 bc | 233.0 cd | 1929 |

[a] Abbreviations as in Table 2; [b] Means with a letter in common are not significantly different ($p < 0.05$).

There were few significant differences in the preferences expressed by females, males and technical staff (Table 11). No differences were observed when the preferences were expressed at flowering, while at maturity, highly significant differences were observed in the preferences expressed by the male farmers, which were reflected in the mean preferences at maturity. The two most preferred entries were the selections by males and technicians, which did not differ from each other, and which, as average preference, did not differ significantly from the mixture of breeding lines (Be_J_Mix3). The two checks, the local landrace (Kalo Male) and the improved variety (Trishuli), were the least preferred together with the mixture of 21 local landraces (Be_J_Mix1), the mixture of 42 landraces (Be_J_Mix2), six breeding lines (Be_J_Mix4) and the female selection (Be_J_F_sel).

**Table 11.** Farmers' and technicians' preferences [b] on seven bean mixtures and two checks grown for three years in Jumla, Nepal (entries 7 to 9 were grown only during the last two years).

| Entry Name [a] | FFL | MFL | TFL | FH | MH | TH | MeanFL | MeanH |
|---|---|---|---|---|---|---|---|---|
| Be_J_Mix1 | 2.93 | 2.71 | 2.78 | 3.07 | 2.71 b | 2.67 | 2.76 | 2.99 cde |
| Be_J_Mix2 | 2.64 | 3.21 | 4.90 | 3.90 | 4.00 b | 4.62 | 3.16 | 3.82 bcd |
| Be_J_Mix3 | 2.99 | 4.36 | 2.41 | 5.47 | 6.26 b | 3.40 | 3.11 | 4.84 abc |
| Be_J_Mix4 | 3.54 | 2.83 | 1.78 | 0.92 | 0.93 b | 1.40 | 2.79 | 0.97 e |
| Kalo Male | 4.14 | 3.48 | 3.77 | 0.79 | 1.98 b | 2.51 | 3.78 | 1.54 de |
| Trishuli | 0.55 | 0.64 | 1.27 | 1.35 | 0.36 b | 0.51 | 0.63 | 0.76 e |
| Be_J_M_sel | 5.66 | 5.50 | 2.02 | 7.33 | 6.67 a | 3.33 | 5.11 | 6.81 a |
| Be_J_F_sel | 5.00 | 3.67 | 1.02 | 3.50 | 2.67 b | 3.33 | 3.95 | 2.92 cde |
| Be_J_T_sel | 4.33 | 5.67 | 3.68 | 6.00 | 6.67 a | 1.33 | 5.00 | 5.81 ab |
| P$_{F(Entries)}$ | 0.373 | 0.106 | 0.547 | 0.262 | 0.0002 | 0.5050 | 0.110 | 0.048 |

[a] Abbreviations as in Table 2; [b] Means with a letter in common are not significantly different ($p < 0.05$); FFL = females' score at flowering; MFL = males' score at flowering; TFL = technical staff score at flowering; FH = females' score at harvesting; MH = males' score at harvesting; TM = technical staff score at harvesting; MeanFL = average score at flowering; MeanH = average score at harvesting.

There was a clear separation between the mixtures based on landraces (Be_J_Mix1 and Be_J_Mix2) or on landraces and breeding lines (Be_J_Mix4) on one side, and the mixture based only on breeding lines (Be_J_Mix3) on the other (Figure 5). The first three had more pods per plant (PodP), a higher 1000 seed weight (TGW), higher yield (GY) and received a high score by technical staffs both at flowering (TFL) and at maturity (TH), while the Be_J_Mix3 tended to have average values for most traits, as indicated by its position nearer the center of the axes. There was an even clearer difference between the three original mixtures (Be_J_Mix1, Be_J_Mix2, and Be_J_Mix4) and the selections made by male farmers (Be_J_M_sel) and technicians (Be_J_T_sel), while the female farmers selection (Be_J_F_sel) was closer to Be_J_Mix3.

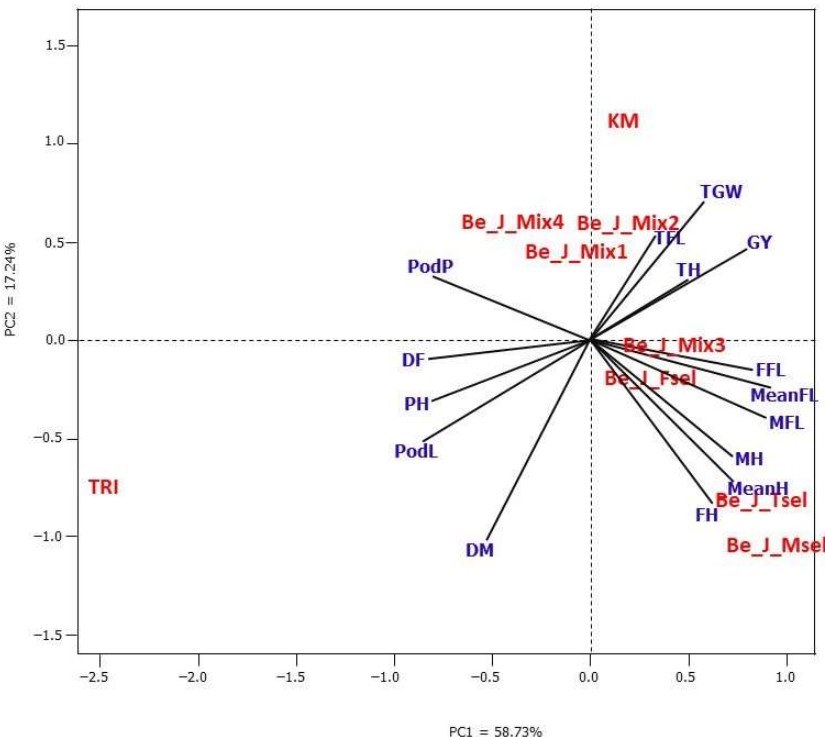

**Figure 5.** Biplot showing the relationships between traits, including agronomic evaluation on seven mixtures and two checks of bean grown in Jumla, Nepal, for three cropping seasons (Be_J_F_sel, Be_J_M_sel, and Be_J_T_sel were grown only during 2020 and 2021 cropping seasons).

There was a positive, although not always very strong, correlation between number of pods per plant (PodP), days to flowering (DF), plant height (PH), pod length (PodL) and days to maturity (DM). The correlation coefficients between days to flowering and number of pods per plant (r = 0.786; *p* < 0.05), days to flowering and pod length (r = 0.820; *p* < 0.01), days to maturity and pod length (r = 0.744; *p* < 0.05), plant height and number of pods per plant (r = 0.761; *p* < 0.05) and plant height and pod length (r = 0.870; *p* < 0.01) were significant (Table S4). These five traits were higher than the mean in Kalo Male used as a check. Technicians, both at flowering (TFL) and at maturity (TH), selected for the lowest expression of these five traits, namely for short plants with fewer and shorter pods and early heading and maturing, and for large seed weight and high grain yield (Figure 5).

## 4. Discussion

We analyzed the 2019–2021 dataset of plant breeding trials conducted with dynamic mixtures in the Lamjung and Jumla districts of Nepal. The field level evidences of rice and beans were derived by conducting the experiments in farmers' fields by using randomized complete block design with three replications. The participatory selection was performed on advanced mixture of landraces, released/registered varieties and breeding lines with the effort of male farmers, female farmers and technicians to produce additional mixtures (entries) and allow more choice to farmers. Similarly, the natural selection acted to produce the mixture with climate resilient agronomic traits suitable to heterogenous environment, where there are fewer options for farmers. We used a total of 56 and 66 rice entries of Lamjung and Jumla, respectively; however, a total of 48 entries of bean were used in Jumla to generate the mixtures with broad genetic base. The germplasm set used in rice for Lamjung and Jumla were totally different to identify and promote location specific varieties.

This research was broadly focused on deciphering agronomic and farmers preference traits of dynamic mixtures by using evolutionary plant breeding as a tool to produce farmers preferred location specific varieties to the hill farmers of Nepal. The objective of this study was to evaluate under farmers' field conditions a number of dynamic mixtures

made with either landraces or with a combination of landraces and improved varieties or breeding lines. We also intended to use the dynamic mixtures as source populations from which the farmers could select the most desirable phenotypes.

The study consisted of three different trials and two crops, rice and beans, as the locations chosen for the study were different and diversity-rich areas for different landraces.

In the high-altitude locations (Jumla), the mixtures, even after a relatively short time, were at the same yield level of the improved variety and significantly higher yielding than the local check. The results of artificial selection seem to indicate a significant divergence from natural selection: for example, in the case of phenology, the farmers' and the technical staff selections were significantly later than some of the mixtures and of the local landraces, suggesting that farmers are taking advantage of the diversity within the mixtures by selecting plants with the most desirable phenology for their local growing conditions. The same applies for plant height: while the mixtures tend to have a plant height similar to the local landraces or slightly taller, the farmers seem to be interested in a crop taller than the landrace and more similar to the improved check. This might be due to their preference for other traits such as straw yield as farmers in hill-farming systems practice an integrated livestock-crop production. Rice straw is the major livestock feed source in the hill farming system of Nepal [32]. In addition, their selection increased the number of tillers significantly, the number of panicles and the length of the panicles. The improvement in these traits, including thousand grain weight, did not result in a detectable yield gain in bean and, actually, only the male selection resulted in a yield comparable with the highest yielding entries. The efficiency of farmers' selection is certainly not a novelty and was reported in other crops [33], including rice [34].

In the rice trials conducted in the location representing the mid-hill region (Lamjung), there were a number of similarities with the results obtained in Jumla, although the material used for the mixtures as well as the checks were different. As already observed in the rice trials in Jumla, also in Lamjung, the mixtures, including the one made only with locally adapted landraces, were among the top yielding. In both locations, artificial selection by farmers and technical staff was effective as it generated material, which was significantly different for a number of traits from the original mixtures that were left to natural selection. Farmers' selection within one of the dynamic evolutionary populations was reported to effectively improve yield and yield stability above those of the original populations across years and locations [23] and such selection accelerates the improvement of traits compared to the original population [35].

In Jumla, the selections were later in heading and maturity, taller, with more tiller and panicles and with longer panicles than that of the mixtures. In some cases, there were significant differences between the selectors: for example, female selections were significantly earlier maturing than either male or technical staff selection, underlining the importance of disaggregating preferences. Despite these changes, only the male selections were included in the top yielding group, which outyielded the local landrace significantly, even if the differences with the selections by the females and the technicians were not significant. The difference in traits preferences of male and female farmers and the preference of male farmers for yield and yield-related traits was also observed in wheat [36].

In discussing these results, we should remember that the dynamic mixtures were left to natural selection for three years and the selections were made in the first year when, given the self-pollinated nature of the crop, the availability of new recombinants expected to be generated by natural outcrossing was limited.

In Lamjung, there was a similar effect, as observed in Jumla, with the selections being later in heading and maturity than mixtures, which were left at the effects of natural selection, although the differences were not as evident, particularly in the case of maturity. At this location, late selections were also taller, even taller than the local check, particularly those made by female and male farmers. However, the major differences with what we observed in Jumla were that although these late selections had a significantly lower number of tillers and panicles, all the three selections were in the top yielding group,

yielding significantly more than the mixture of five improved cultivars (Ri_La_Mix3) and not significantly different from Gaure.

In general, farmers' and technical staff' preferences went for the highest yielding: the Mix 1, which in both locations was based on local landraces, was both the highest yielding mixture and one with a high preference score. However, a high preference score was also given to Jumli Marshi and Gaure, the two local checks used in Jumla and Lamjung, respectively, although only Gaure was in the high yielding group.

While in the case of rice, at least some of the mixtures under the sole effect of natural selection and some of the material selected by farmers and technical staff performed as well as the improved varieties used as check like, for example, in the case of grain yield, in the case of beans, the beneficial effect of the mixtures, either under natural or artificial selection, was much less evident. Differences between entries were masked by large entries x years interactions and were detected only for plant height, pod length and thousand grain weight.

In beans, the two checks were the tallest entries and both the mixtures and the selections were shorter without any evidence that the selection made any difference. Similarly, for pod length, none of the mixtures matched the long pods of Trishuli; however, there was some improvement over the local check. There was a large difference in thousand grain weight between the two checks, with the local Kalo Male characterized by much heavier seed than Trishuli. The mixtures and the selections seem to indicate changes in the direction of the seed weight of Kalo Male.

Despite the yields varying from 955 to 2037 kg ha$^{-1}$, the differences were not significant (F = 1.65 with 8 and 13 df, $p$ = 0.202). The F test for the entries was performed against the entries x years interaction which was significant ($p$ = 0.021). By inspecting the entry x years means, it became obvious that Trishuli (the improved check) was the problem, as it yielded 2030 kg ha$^{-1}$ in 2019, 987 kg ha$^{-1}$ in 2020 and 574 kg ha$^{-1}$ in 2021 and this affected the difference between years that was highly significant ($p$ < 0.00001). Trishuli is adapted to favorable conditions and its yield instability is typical of such varieties when grown in low input conditions [37]. When Trishuli was removed from the analysis, the entry x years interaction was no longer significant and the differences between entries remained not significant. There was also limited evidence for differences in the preference score. Only at harvesting, males expressed a strong preference for their own selections and for those made by the technical staff, which, in terms of the measured traits, did not show any particular association, as suggested by the biplot in Figure 5.

The current seed regulation of Nepal does not allow to register mixtures and evolutionary populations as a common variety and be accepted by the market in Nepal. Particularly, the varietal registration guidelines of Nepal do not provide ample space to register evolutionary populations and dynamic mixtures which hinders farmers to produce and sell those varieties. However, the evidence generated from this work indicates that they can represent an excellent genetic material for the institutions responsible for plant breeding to use in a decentralized-participatory breeding program in future. Similarly, the breeding materials generated from this research will provide further choice of germplasm to farmers and contribute to sustainably increase in yield to the hill farmers of Nepal.

## 5. Conclusions

Overall, dynamic mixtures are a reliable source of germplasm for increase in yield and agronomic preferences for low input and hill farming systems of Nepal. The most important finding of this work was that, at least in rice, one cycle of visual selection by farmers and technicians significantly increases agronomic and farmers preference traits in the dynamic mixtures. The integration of natural and artificial selection seems highly effective to increase in yield of dynamic mixture of rice varieties generated from landraces, improved varieties and breeding lines; however, there was no clear evidence in beans. The performance of some mixtures and selections from the mixtures gave a grain yield comparable to the improved check and higher than the local check, showing superior

performance of these mixtures. In bean, because of a high entries x years interaction, we found less evidence of changes due to selection within mixtures, although some of them received a high preference score. Overall, the dynamic mixtures appear as the reliable sources for sustainable increase in yield in the low input and hill farming system of Nepal.

**Supplementary Materials:** The following supporting information can be downloaded at: https://www.mdpi.com/article/10.3390/d15050660/s1.

**Author Contributions:** Conceptualization, S.C. and S.G.; methodology, S.C., S.G., S.P.N. and D.K.A.; software, S.C.; validation, S.C., S.G., S.P.N. and D.K.A.; formal analysis, S.C.; investigation, S.C., S.G., S.P.N. and D.K.A.; resources, B.K.J. and D.G.; data curation, S.C., S.G., S.P.N. and D.K.A.; writing—original draft preparation, S.C.; writing—review and editing, S.C., S.G., S.P.N., D.K.A., B.K.J., D.K.A., D.G., D.I.J. and D.K.M.; visualization, S.C. and S.P.N.; supervision, S.P.N., B.K.J., D.K.A., K.H.G., A.K. and D.K.M.; project administration, D.G., D.I.J. and D.K.M.; funding acquisition, D.I.J. All authors have read and agreed to the published version of the manuscript.

**Funding:** This research was funded by the International Fund for Agricultural Development (IFAD), grant number 2000001629 and The APC was funded by EPB-IFAD project.

**Institutional Review Board Statement:** Not applicable.

**Data Availability Statement:** Data are available from the senior author on reasonable request.

**Acknowledgments:** The authors acknowledge IFAD's financial support and the help of a number of technical staff and of female and male farmers in Nepal.

**Conflicts of Interest:** The authors declare no conflict of interest.

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
