# Peer review of "Farmers’ Preferences and Agronomic Evaluation of Dynamic Mixtures of Rice and Bean in Nepal"

_diversity, doi:10.3390/d15050660_

Round 1
Reviewer 1 Report
1. Lin e45, ithas add spaces.
2. The instruction needs to have a sentence to point out where the predecessors have not studied enough, and what is the entry point of this research.
3. Lin 279, 592 add a full stop.
4. In the discussion, increase the discussion with other studies, with references.
5. Line 595, remove a comma.
6. The uppercase and lowercase letters in the reference should be consistent.
Author Response
The comments and suggestions were addressed carefully. See attached document

Reviewer 2 Report
Review of the manuscript titled: “Farmers’ Preferences and Agronomic Evaluation of Dynamic Mixtures of Rice and Bean in Nepal”.
The study was well designed and represents a large effort of comparing mixtures, varieties and farmers selections of two crops at three sites. The methodology is very relevant, the study results are significant and contribute to our knowledge on crops diversity and evolutionary breeding approaches. It is certainly worth publishing. The main challenge of the paper is how to efficiently incorporate the results from three large and rather sophisticated trials into one document. The authors try to present everything and for this reason the paper is a little “heavy” and not easy to follow especially considering many abbreviations. Perhaps, not all data collected need to be presented and discussed but rather focus on the main study outcomes. There are also few concrete suggestions.
11. The literature review is one-sided in favor of EVP and can be more balanced citing cases when EVP was not as efficient as regular breeding approaches.
22. Table 1 has almost the same information in two main columns. It can be restructured to provide more details on the mixtures composition or removed altogether.
33. Table 3 is cited in the text before Table 2. They need to be replaced or cited accordingly.
44. “The list of entries and the composition of the different mixtures of rice and bean are reported in table 4.” Table 4 is devoted to ANOVA results and contains no such list.
55. There are too many abbreviations, both in texts and in tables. The abbreviations for tables are listed for each table – no “space saving”. Why to abbreviate the names of check varieties, Gaure, for instance. There is no need to abbreviate traits unless they are composed of three words. The selection groups also do not require abbreviations. Perhaps only mixtures can be abbreviated. Example: “Grain yield was negatively correlated with Tsel and not correlated with either Ri_J_F_sel or Ri_J_M_sel.” The sentence meaning is not clear and can be easily explained without abbreviations by saying that selections by technicians were lower yielding as opposed to selection by males and females.
66. The biplots are based on mean values across years. May be it is fine if the seasons were similar – the first section of results shall address this issue. If the seasons are very different – then perhaps the biplots do not reflect the real situation and some other way of presenting the traits relationship is needed.
77. Lines 504-510 and Figure 5 do not belong to discussion. Rather this shall be the first introductory part of the results to explain the differences between the years and sites from agronomic and biological perspective.
88. Lines 580-592 – repetition of results.
99. Discussions needs better focus on the key broad study outcomes and their application for crops production in the highlands of Nepal and probably beyond. EPB so frequently mentioned in introduction, not mentioned in discussion even a single time. How the study results contribute to perspectives of EPB.
Author Response
The comments and suggestions addressed in the attached document

Reviewer 3 Report
The study sought to demonstrate the male, female and technical point of views regarding some production characteristics of Rice and Bean in Nepal, but there were many small and large difficulties in describing the data, as I will clarify below:
The introduction is very long and has entire paragraphs that can be removed without prejudice to the text, as is the case between lines 72-74.
Lines 75-81. This text would be more cohesive if joined to the paragraph between lines 44-60 and not as an isolated paragraph.
Image 1 quality is compromised, especially the distance caption which is completely unreadable.
Table one could be transferred to supplementary material as it does not bring much information to the reader, rather it is supplementary information.
Line 129: “scent to obtain disease tolerance” confuse, explain better.
How was stem and leaf color measured? I did not find the description of this methodology.
How was Aromatic and non-aromatic traits measured?
Table 4 is called in the text before table 3. Double-check, pl.
Lines 186-187. “Grain yield was adjusted for moisture at 13% and 10%”. How was it done? Describe in the methodology.
Lines 209-209, the text is confusing, please rewrite better.
The material is methods is very confusing and long. Needs to be rewritten.
Lines 301-304. “The selection done by the technician had the largest TGW, significantly higher than the improved check but not significantly different from the male and female selections and from the local check.”. Confuse text, improve explanation.
The results stress a lot of technical differences, female and male, a fact that confuses the reader and masks the main data. I suggest improving this fragmented text.
Lines 342-348 describe correlations between response variables, but it is not stated in which table or figure these correlations are presented. If the correlations are being insipiently shown in Figure 2, I inform you that vectors in negative directions do not necessarily show negative correlations (because the strength that this happens may not be significant at P<0.05), the most correct, in this case, would be to show a Pearson correlation table with its p value or the table of scores that the PCA generates at the end of the graph.
I would like to see the data in the results, not just saying greater, lesser or significant. This text has to be improved.
The discussion is short and does not encompass all the data collected and presented in the results section.
As presented in Instructions for Authors, in the discussion section the “Authors should discuss the results and how they can be interpreted in perspective of previous studies and of the working hypotheses. The findings and their implications should be discussed in the broadest possible context and limitations of the work highlighted. Future research directions may also be mentioned”, but the discussion with previous works is very deficient and I was unable to envision the Future research directions. I suggest rewriting this section using the Diversity – MDP, Instructions for Authors.
The discussion seems to me more like an extension of the results section than a discussion per se.
The conclusions do not support the data presented, it is far from what I was able to read and understand from the work.
Author Response
The comments and suggestions were included in the attached document.

Round 2
Reviewer 2 Report
The authors did good job with the revisions and the paper can now be published.
Reviewer 3 Report
The present version of the manuscript was improved when compared to the original version. Some minor consideration is highlighted on the paper in yellow

Author Response
The request was addressed
